# Reinforcement Learning Based Multipath QUIC Scheduler for Multimedia Streaming

**DOI:** 10.3390/s22176333

**Published:** 2022-08-23

**Authors:** Seunghwa Lee, Joon Yoo

**Affiliations:** School of Computing, Gachon University, 1342 Seongnam-daero, Sujeong-gu, Seongnam-si 13120, Korea

**Keywords:** multipath QUIC, DQN, reinforcement learning, scheduler

## Abstract

With the recent advances in computing devices such as smartphones and laptops, most devices are equipped with multiple network interfaces such as cellular, Wi-Fi, and Ethernet. Multipath TCP (MPTCP) has been the de facto standard for utilizing multipaths, and Multipath QUIC (MPQUIC), which is an extension of the Quick UDP Internet Connections (QUIC) protocol, has become a promising replacement due to its various advantages. The multipath scheduler, which determines the path to which each packet should be transmitted, is a key function that affects the multipath transport performance. For example, the default minRTT scheduler typically achieves good throughput, while the redundant scheduler gains low latency. While the legacy schedulers may generally give a desirable performance in some environments, however, each application renders different requirements. For example, Web applications target low latency, while video streaming applications require low jitter and high video quality. In this paper, we propose a novel MPQUIC scheduler based on deep reinforcement learning using the Deep Q-Network (DQN) that enhances the quality of multimedia streaming. Our proposal first takes into account both delay and throughput as a reward for reinforcement learning to achieve a low video chunk download time. Second, we propose a chunk manager that informs the scheduler of the video chunk information, and we also tune the learning parameters to explore new random actions adequately. Finally, we implement our new scheduler on the Linux kernel and give results using the Mininet experiments. The evaluation results show that our proposal outperforms legacy schedulers by at least 20%.

## 1. Introduction

Nowadays, most computing devices, including laptops and smartphones, can exploit multiple network interfaces, e.g., 4G/5G and Wi-Fi, for data communication. Most Internet applications, however, still use TCP as the transport protocol [1], whereas the original TCP can only use a single network interface. Various multipath protocols [2,3,4,5] were developed to take full advantage of multiple network interfaces. In particular, Multipath TCP (MPTCP) [3,4] was proposed to utilize multiple network interfaces at the transport layer. In MPTCP, the kernel creates a subflow for each network interface, and when new data is transmitted from the TCP socket, the kernel transmits a packet to one of the network interfaces, determined by the MPTCP scheduler. All these operations are done transparently to the application so that the applications do not need any modifications. However, the main challenge of MPTCP is that the server and client kernels must be modified, hindering its popularity and wide deployment.

Meanwhile, Multipath QUIC (MPQUIC) [5] has been proposed as an alternate to the MPTCP. It is an extension of the Quick UDP Internet Connections (QUIC) proposed by Google [6], which is a TCP-like transport protocol implemented at the application layer. The main advantages of MPQUIC are as follows. First, since MPQUIC is implemented at the application layer, it can be easily distributed to users through simple application updates. On the other hand, MPTCP needed kernel updates. Second, it can be custom-tailored to meet the requirements of a specific application. On the other hand, since MPTCP is implemented in the kernel, it must meet the general requirements of various applications such as web browsing, large file transfer, video streaming, and chatting. Therefore, MPTCP scheduling algorithms were developed to achieve general goals such as latency and bandwidth [7,8]. Third, some useful information provided by the application layer can be naturally utilized in QUIC. For example, in a video streaming system, information such as bit rate, video chunk size, and video buffer can be obtained for the MPQUIC scheduler.

This paper focuses on video streaming applications such as Dynamic Adaptive Streaming over HTTP (DASH) [9]. To the best of our knowledge, existing studies of the multipath scheduler have not been fully optimized for the video streaming environment. The main reason is that most schedulers attempt to optimize either *latency* or *throughput*. The default multipath scheduler, MinRTT [3] first selects an available path with the smallest RTT, but once that path becomes full, it opportunistically utilizes all other paths. Therefore, when using large file transfer, minRTT can achieve high throughput but is not optimized in terms of latency since the large RTT paths may increase the overall application delay. The redundant scheduler [10], however, reduces latency by transmitting duplicate packets on multiple paths, but the final throughput is inevitably lowered because it generates a lot of redundant data.

In this paper, we propose an MPQUIC scheduler to improve the performance of video streaming via a novel deep reinforcement learning (RL) algorithm [11] that uses a Deep Q-Network (DQN) [12]. The reason why we use reinforcement learning is twofold. (i) The multipath scheduler can only obtain some of the network parameter information, e.g., cwnd or RTT, not the global network information. However, the optimal scheduling algorithm may differ depending on the global network parameters. In this case, reinforcement learning has the advantage of being able to learn on its own, even when some of the network parameters are generally unknown. (ii) In multimedia streaming such as DASH, there exists ample opportunity to explore new actions when there are enough video chunks in the application buffer.

The main contributions of our paper are as follows.

**Chunk download time reduction.** The existing reinforcement learning scheduler adopts either throughput or RTT as a reward. However, in the DASH environment, if only one is optimized, performance may be degraded. A new reward is required because the metric to be selected is different according to the transfer progress. In this paper, we solve this problem using two rewards. The episode is first defined as the transmission of one video chunk, and at the end of the episode, a reward is determined as inversely proportional to the download time. This helps the agent to learn the appropriate policy for each episode. However, since many actions can occur in one episode, the agent may not learn well with the reward of the episode alone. To solve this, we implemented each action within the episode to obtain a partial reward proportional to the throughput.**Chunk manager design.** The chunk manager is used by the scheduler to accurately detect the transmission state of the video chunk. For example, the scheduler can be informed of how much of the video chunk has been successfully received by the client. However, unlike MPTCP, it is difficult to accurately determine the transmission state in MPQUIC due to the MPQUIC server-client structure. Therefore, we present these problems and solutions.**Reinforcement learning exploitation vs. exploration.** In reinforcement learning, it is important to use learned policies (exploitation), but it is also very important to explore new actions and update values (exploration) as a given environment can change. However, selecting a new action can instantly degrade performance significantly and lead to poor user experience, such as rebuffering. Therefore, in this paper, the buffer level information of the client is utilized before performing the exploration. By performing a random action only when there are enough video data in the buffer, new actions can be explored without affecting the client’s performance. At the same time, when estimating the download speed for the client to determine the available bitrate, the video chunk that performed the random action is excluded from the bitrate calculation. This avoids underestimating the network.**Multithreading-based RL scheduler implementation.** The DQN generally conducts prediction, action, and learning in order. When we first implemented the reinforcement learning (RL) scheduler, its performance degraded. The main reason is that the MPQUIC scheduler requires a real-time process, but conducting the learning and scheduling together induces too much CPU overhead. We solved this problem by separating the learning thread from the scheduler thread. In this case, another challenge is that the learning thread and the MPQUIC main thread both access the same policy variable. This may cause a fatal error in the application. To solve this issue, we divided the DQN’s online policy, target policy, and prediction policy and periodically synchronized them. Synchronization can be performed in a very short time, which does not affect the performance of the scheduler.

The remainder of the paper is structured as follows. Section 2 introduces background knowledge and related work. Section 3 describes the new scheduler design using reinforcement learning and Section 5 describes the experimental setup and the evaluation results. Finally, Section 7 concludes this paper.

## 2. Related Work

In this section, we present the recent multipath protocols, e.g., MPTCP and MPQUIC, and then we give the optimal scheduling problem in DASH. Finally, we present the recent multipath scheduling protocols.

### 2.1. Multipath Transport Protocols

Multipath TCP [3,13,14,15] enables the user application to utilize multiple network interfaces for better performance and reliability. When a new TCP connection is established, a TCP socket called *subflow* is created in the kernel for each network interface. The kernel hides the existence of the subflow to the application by providing a *meta socket*. That is, the data stream received through the meta socket is distributed from the kernel to the subflows. The advantage is that existing TCP socket programs can be used without any modifications while enjoying the benefit of MPTCP.

The MPTCP scheduler is a crucial part of MPTCP; When a packet is sent from the application to the MPTCP layer, the MPTCP scheduler decides which path to send the data. The default scheduler is called *MinRTT*. MinRTT finds subflows that have available congestion window space. Among them, the data are transmitted to the subflow with the minimum RTT, hence MinRTT. This scheduler aims to increase the throughput by filling the congestion window (CWND) of all subflows.

Despite the aforementioned advantages of multipath, in some cases, MPTCP shows worse performance compared with single-path TCP [16]. The main cause is *head of line (HOL) blocking* [8,17]. HOL happens when data is sent to multiple subflows, the packets may not arrive in order at the receiver side because the network quality of each path is different. However, TCP guarantees that the data packets are delivered in good order. Packets that do not arrive in order are not delivered to the application, and kernel memory has to wait for packets that fill the missing gaps. Here, the receive buffer becomes larger due to waiting. In this case, the sender recognizes that the receive buffer is insufficient and responds by lowering the throughput. Various schedulers have been proposed [8,17] to solve this problem, and we discuss them in Section 2.3.

Multipath QUIC (MPQUIC) [5] is a multipath extension of QUIC [6]. In MPQUIC, the sender can transmit multiple files simultaneously by connecting multiple files to each *stream*. For this purpose, MPQUIC uses a frame structure so that it allows multiple frames to be dynamically appended to one packet. A frame containing stream data is added to the packet as *Stream Frame*. A frame containing ACK is combined into a packet as *ACK Frame*.

### 2.2. Dynamic Adaptive Streaming over HTTP (DASH)

DASH [9] is one of the most widely used adaptive bitrate streaming algorithms in video streaming. First, the Web server divides the video file into small video files, called *chunks*, and stores them using several different video bitrates. The client then can request chunks to the Web server by using an HTTP request. Here, the client may request a higher or lower bitrate chunk depending on the estimated network conditions.

In most cases, throughput and application latency can be trade-offs when scheduling a packet. Therefore, there are many schedulers that try to optimize either one of them. However, improving either may not yield the best performance in DASH scenarios. Let us give an example network configuration as shown in Table 1. There are two paths and the first path has a latency of 5 ms when the packet is transmitted, and the second path has a latency of 50 ms. CWND means the congestion window and the number of packets that can be sent before receiving the ACK. That is, path 1 can transmit 3 packets at a time. In this situation, we will explain a scheduling method that can quickly transmit a chunk with a size of 100 packets.

Figure 1 shows the optimal strategy for scheduling in a given environment. When the scenario starts, it may be the best strategy to use all available paths, as shown in Figure 1a. However, this strategy may not be a good strategy when the chunk is almost finished. It is assumed that packets 96, 97, 98, and 99 remain. If 3 packets (96, 97, 98) are allocated to Path 1 and one packet (99) is allocated to Path 2, chunk transmission will be completed after at least 50 ms. The reason is that it has to wait until packet 99 arrives on Path 2. In this case, packets 96, 97, and 98 will be transmitted within 5 ms and fast Path will wait 45 ms without transmitting anything.

On the other hand, if we change the scheduling strategy, assign packets 96, 97, and 98 to Path 1 and do not use Path 2, as shown in Figure 1b, different results may occur. There will be space in the cwnd of Path 1 within 10 ms (one-way delay × 2). After that, if packet 99 is assigned as Path 1, chunk transmission will be completed within 15 ms in total. This is shown in Figure 1c. This is a scheduling strategy that can complete the transmission of chunks faster than the existing method using all paths by at least 35 ms.

### 2.3. Multipath Scheduling Algorithms

Earliest Completion First (ECF) is a scheduling algorithm that estimates the packet arrival time for each path before scheduling the packets so that packets can arrive sequentially [17]. Therefore, there is an advantage that out-of-order packets do not occur in a situation where there is no packet loss or jitter.

Blocking estimation-based MPTCP scheduler for heterogeneous networks (BLEST) is a scheduling algorithm that focuses on the send window [8]. Send window is the window that the sender has. When sending a packet, there must be free space in the send window. The reason for storing packets in the send window is to prepare for retransmission due to packet loss.

Peekaboo is an MPQUIC scheduler that uses reinforcement learning [18]. In Peekaboo, the fast path is used first whenever it is available, and then the Peekaboo determines whether it will wait or send the packet to the slow path. It uses reinforcement learning to determine this by using three states (CWND, in-flight packet, and send window) and using throughput as the reward.

## 3. Reinforcement Learning Based Multipath QUIC Scheduler

Figure 2 shows the deep-Q network (DQN) used for the reinforcement learning used in our paper. First, the *MPQUIC scheduler agent*, shown on the left side of the figure, determines the action. The agent gets *state* input, such as RTT, CWND, transmission state of the chunk, and send window (SWND). In DQN, the Q-Table (policy) is implemented as a deep neural network (DNN). Therefore, the state is placed in the input layer of the DNN and the output of the DNN is given as *action*. Finally, the action that affects the environment and as this process is repeated, the state (previous), the action, the reward, and the observation (state) are stored in the *replay buffer*. The online policy is learned in a separate thread using the replay buffer. After training, we optionally update the parameters of the target policy and the policy within the agent. The reason for not regularly updating the target policy is due to the double DQN [19].

The remainder of the section is organized as follows. In Section 3.1 we describe the main components of reinforcement learning: episode, action, state, reward, then in Section 3.2 we describe the main algorithm.

### 3.1. Reinforcement Learning Components

#### 3.1.1. Episode

For reinforcement learning, *Episode* is defined as transmission length of a chunk. In other words, if the server sends 10 chunks, the agent will experience 10 episodes. As shown in Figure 3, multiple *Actions* can be included in Episode. These actions can be scheduled *per packet* [18] or can be scheduled for every predetermined *interval* [20].

In this paper, we use the latter interval policy as shown in Figure 3, and the interval is predetermined as 50 ms. That is, the scheduling decision is made every 50 ms and is maintained throughout the 50 ms interval. We state two reasons for selecting this policy. First, when we execute the scheduling based on the reinforcement learning algorithm for every packet, the CPU overhead becomes large enough to affect the chunk download time, especially in high-speed networks. We have conducted a few experiments that showed evidence of this phenomenon, but we omit this due to space limitations. The second reason is the reward. It is advantageous for the reinforcement learning algorithm to know the reward for each action taken as soon as possible. However, due to the latency of the network, it takes time to know whether or not a scheduled packet has been successfully transmitted. Due to this delay between action and reward, there can exist multiple other actions in-between an action and a reward; thus, the causal relationship between action and reward becomes weak. We analyzed that this increases the difficulty of reinforcement learning.

However, determining the scheduling policy once every 50 ms degrades the granularity of the scheduling policy. Therefore, this problem is solved by creating an action that performs a different scheduling method over time.

#### 3.1.2. Action

An *Action* means the scheduling policy to be executed for 50 ms. There are seven types of Action as shown below:**Probabilistic scheduling (3 types).** When this action is selected, Path 1 is selected according to a predetermined probability; otherwise, Path 2 is selected. The selection probabilities are 0.25, 0.5, and 0.75, and each probability represents a different action. If the CWND of the selected path is full, it waits without sending the data to any of the paths. This scheduling method has the advantage of the capability of limiting the use of an undesired path.**Only fast path.** As explained in Section 2.1, in some cases, it may sometimes be good to rule out the use of the slow path. Therefore, an action has been added to select only the fast path based on the RTT.**MinRTT.** This action executes the same as the minRTT scheduler. This action will exploit all paths’ CWND, and thus is expected to show good throughput.**Path scheduling over time (2 types).** In some cases, 50 ms may be too coarse, in which case we subdivide the scheduling action. The MinRTT scheduling method is used first, and the fast path scheduling method is used after a certain period of time. In one action, we use minRTT for 15 ms, then for the remaining 35 ms use fast path. The second action uses 30 ms for minRTT, then the remaining 20 ms for the fast path.

#### 3.1.3. State

The data collection is done on two occasions. First, when scheduling is conducted, the data tuple [State (50 ms before), Action (50 ms before), Reward (50 ms during), State (current)] is collected every 50 ms, provided that the chunk download is not completed. Second, when the chunk download (or episode) is completed, the data tuple [State (50 ms before), Action (50 ms before), Reward (episode), and State (current)] is collected. The data training algorithm is divided from the scheduler function by the GoRoutine (Go language thread).

We use a total of 10 *States*: 6 features that can be obtained from 2 paths and 4 features that can be obtained from the QUIC session. Features that can be obtained from the paths are *Smoothed RTT*, *Congestion window*, and *Inflight bytes*, which represent the latency of the path, the maximum size that can be transmitted, and the number of bytes being transmitted, respectively. Other features include *the size of the send window* and *elapsed time for download*. There are also features related to chunks. One of them is *the size of the remaining chunks that need to be transmitted*. This feature helps the scheduler check the amount of data remaining and to use a policy that limits the slow path as this value decreases. The other is *the size of the total chunk*-the size of the whole chunk received by the client. This value is generated based on ACK.

#### 3.1.4. Reward

First, the *reward* that is basically used is throughput. This value is determined by using how many packets arrive as ACKs during the 50 ms of the scheduling action. This reward will help the agent learn to send as many packets as possible with efficiency.

In addition to this, this paper uses episode reward. This is a reward provided to the agent when the chunk download is complete. In this paper, the episode reward is inversely proportional to chunk download time and proportional to the chunk size. This will help optimize the transfer of chunks. Therefore, the scheduler will try to optimize the throughput during the earlier part of the episode and perform actions to reduce the overall download time by appropriately limiting the two paths as the chunk transmission is finished.

### 3.2. Learning Algorithm

We use the Deep Q-Network (DQN) [12] for the reinforcement learning algorithm, as shown in Algorithm 1. In this algorithm, the policy represents the value of each action that can be obtained from a state. So, the state is given as an input to the policy, and the [1*action] vector returns a value as an output. The *replay buffer* records the actions performed by reinforcement learning and the rewards obtained, and is represented as a set of tuples (=events), of the vector [State, Action, Reward, Observation (changed State)].

As shown in line 2 of Algorithm 1, the stored events are randomly extracted and used every time the learning is performed. Lines 3 to 12 are the process of learning the policy. The policy input is the state. Since the output is a value, the process of calculating the value is performed. Basically, the value includes the reward in the current state. So, in line 4, the current reward is entered as a default value in qUpdate.

On line 5, we check whether the state indicates the end of the episode. If it is not the end, future rewards should be considered together. The future reward calculation uses the target policy. It utilizes the state value changed due to the event. The changed state is also called the observation state. DQN utilizes the learned target policy to calculate future rewards in line 6. The return value of the policy is a vector. Each element of the vector represents the predicted value that can be obtained when each action is performed. At this time, the agent will select the action with the highest value. Therefore, the highest value becomes the future reward in line 7. Finally, the *qValue* becomes the current reward + the highest value extracted in the previous process.

*Gamma* limits future rewards. Without this gamma, the *qValue* will not converge. In this process, *x* and *y* are extracted to learn the policy. *x* is the current state, and *y* is a value considering the future.

In line 14, we finally proceed with learning. Note that this point is where *OnlinePolicy* is learned, not *TargetPolicy*. The *TargetPolicy* is updated at regular intervals in line 18. This reduces the problem of overfitting [19].
**Algorithm 1** Learning Policy**Require:** TargetPolicy, OnlinePolicy, ReplayBuffer, batchSize = 20, gamma = 0.5, step = 50  1:  *X* = [], *y* = []  2:  events = Sampling(ReplayBuffer, batchSize, ‘Random’)  3:  **for all** events **do**  4:         qUpdate = event.Reward  5:         **if** event.dDone is False **then**  6:            qValues = TargetPolicy.Predict(event.Observation)  7:            maxqValue = max(qValues)  8:            qUpdate = qUpdate + gamma * maxqValue  9:         **end if**10:      *X*.append(event.State)11:      *y*.append(qUpdate)12: **end for**13: 14: OnlinePolicy.Learn(*X*, *y*)15: 16: **if** agent.ActionStep is greater than Step **then**17:        agent.ActionStep = agent.ActionStep-targetStep18:        Synchronize(to = TargetPolicy, from = OnlinePolicy)19: **end if**

### 3.3. Hyperparameters

In this Section, the explain the *hyperparameters* used in the scheduler.

**Gamma.***Gamma* denotes the discount factor that the agent considers when calculating the policy. A value between 0 and 1 should be selected. If this value is large, it means that the agent considers the result of the distant future when deciding the action. But if too large a value is selected, the policy value may not converge, and the agent may make the wrong choice because the future reward of each action is similar. If this value is too small, the episode will not be considered and a policy of optimizing only throughput will be learned piecemeal. In this paper, the optimal value of 0.5 was used through experiments.

**Update target step.** There are two types of policies: *Target Policy* and *Online Policy*. The *target policy* synchronizes the *online policy* once at a set interval. This can reduce the overfitting of the model [19]. However, if the step is set to a value that is too large, underfitting may occur. Therefore, We set this value to 50. This means that every time 50 actions are performed, the target policy is updated.

**Batch size, Buffer size.** These are used for the replay buffer. *Batch size* determines how much data to store and *buffer size* defines how much data to use for learning at a time. In this paper, there is no memory limit for the events that are stored and 20 events are sampled.

**Epsilon.** Since the agent does not have information about the environment in the first place, it must perform a random action. At this time, *Epsilon* is the probability that reinforcement learning will perform a random action. This value starts at 1 and the rate of decrement and the lowest value can be adjusted. Each time we perform an action, we multiply the value by 0.997 to lower the value.

## 4. Chunk Manager

The *chunk manager* helps the reinforcement learning scheduler to be informed of the chunk information. It transmits the chunk information to the scheduler and records which part of the chunk has been delivered successfully. Moreover, it extracts the client’s chunk buffer status from the HTTP header and delivers to the agent whether a random action should be performed.

Another important role of the chunk manager is to measure the chunk download times to calculate the reward. Chunk download time is the difference between the time when all the ACKs for the entire chunk are received and the time when transmission starts. We then subtract 0.5×RTT and ACKdelay from this value. 0.5×RTT is the time it takes for the last ACK received to be transmitted from the client to the server. ACKDelay is either caused by CPU Time for ACK processing or by the use of *delayed ACKs* (Delayed ACK is used to reduce the number of ACKs sent for network efficiency. However, this may cause an overestimation of RTTs.) [21].

The most important part of the chunk manager is to understand the transmission status of chunks through ACK. If the transmission status can be accurately identified, the server can use an appropriate scheduling policy and obtain the episode reward that is inversely proportional to the download time. However, it is difficult to implement directly using ACK on MPQUIC. The reason is that the first out of order packet, or packet loss, may occur. The second reason is that each path has an independent ACK sequence. In other words, suppose that the last part of the chunk is assigned as Path A. And the server has received all ACKs of Path A. However, the chunk download may not have been completed. The reason is that the middle part of the chunk may be in transit with a different path. In particular, QUIC uses a structure that inserts stream frames into packets. This means that there can be nonconsecutive stream frames in a single packet. In particular, part of the stream contained in each packet may be duplicated. In these issues, sophisticated algorithms are needed to know which parts of a chunk have been transmitted. Therefore, for this part, a module was developed to manage the chunk upload status as a linked list and to identify which part was not transmitted.

Figure 4 explains the overall structure of the operation system. The agent located in the center of the figure delivers the policy to be executed for 50 ms to the scheduler. The scheduler decides which path to be delivered for each packet by using the policy. When an ACK arrives for the transmitted packet, it is transferred to the chunk manager and throughput monitor module, and the processed information is again inputted into the agent. When 50 ms has elapsed, the Agent binds the past state, the action performed, the reward obtained, and the current state and stores it in the replay buffer.

The learning process uses GoRoutine to prevent scheduling performance degradation. This is similar to threads in the Go language.

## 5. Evaluation

This section presents the results of the evaluation of our proposal compared to legacy schedulers. We first give the evaluation setup, then present results on a static network and then on network with packet loss. We then study the performance of the chunk manager and show the performance for adaptive video bitrates. We compare our reinforcement learning (**RL**) scheduler with the following schedulers. **RTT** is the default MinRTT [3] scheduler used in both MPTCP and MPQUIC. We give details on **ECF** [17], **BLEST** [8], and **Peekaboo** [18] in Section 2.3.

### 5.1. Evaluation Setup

We developed and evaluated the MPQUIC scheduler on the Ubuntu 18 Linux kernel. Initially, our scheduler was implemented using the MPQUIC open source proposed by the original authors [5]. ECF [17], BLEST [8], and Peekaboo [18] schedulers were also added with reference to [18]. For BLEST and ECF, some modifications have been made to work well in DASH scenarios.

We use the Mininet open source to implement a virtual network [22]. This supports running programs in a virtual network with the python API or the comment command. A DASH Web server using QUIC is implemented through Caddy [23]. After that, we got help from MAppLE [24] to create the overall framework. The MPQUIC version used by MAppLE and our project is different, so some modifications are required. In MAppLE, Caddy was used as a Web server for DASH streaming. It is an open-source web server that can be easily integrated with QUIC. AStream is a DASH performance measurement program through simulation that is publicly available on GitHub. However, as it is made in Python, quic-proxy is applied to use QUIC. The quick-proxy is a module that executes QUIC written in Go language for HTTP Requests and returns the result.

In all experiments, the segment time is fixed to 1 s, and the length of the video is 300 s, so there are 300 chunks.

### 5.2. Network with No Loss and Diverse RTT Paths

The network environment for the first experiment was configured as shown in Figure 5. There is a fast path with a bandwidth of 20 Mbps and a delay of 5 ms, and a slow path of 15 Mbps and 77 ms. Note that there is no packet loss in this environment.

The result of this is shown in Figure 6. The graph shows the download time of the chunk. Our scheduler performs a random action when there are enough chunks in the client’s buffer. This can temporarily increase chunk download times. However, when playing a new video or searching for time, a random action is not performed. Therefore, the experimental result does not include the chunk on which the random action is performed. However, it is necessary to match the number of chunks included in the graph boxplot between schedulers. So even for the scheduler without the random action function, some chunks were excluded from the result.

At the beginning of the chunk download, the performance of the scheduler is not good because there is no information about the environment. However, looking at the graph Figure 6b for the chunk after 150 s, it can be seen that the performance of the RL scheduler is improved. ECF is a scheduler with similar performance to a reinforcement learning scheduler. This is because ECF can solve the chunk download problem presented in Section 2.2 by estimating the packet arrival time. In particular, it seems to have produced good results because there is no estimation error because there is no packet loss or jitter.

Meanwhile, Peekaboo performed poorly compared to other schedulers. The main reason is that it is a scheduler that optimizes throughput, not the file transfer completion time. First, it takes time to estimate throughput (=average throughput of packets measured over a period of time), but since chunk download is usually within several hundred KB, it is finished very quickly. Therefore, there is often no data to send. At this time, if the throughput is estimated, it has a value close to 0. That is, there is a problem in which the wrong reward is given to the agent. Our scheduler prevented the situation where the throughput was measured incorrectly by not recording this situation in the reinforcement learning memory used in Peekaboo. The second reason is Peekaboo’s stochastic adjustment strategy. Peekaboo reverses the action with low probability. This is to update the value, but if the slow path is used incorrectly in the scenario, performance may be degraded.

Figure 7 studies the scheduler performance change over time. Chunks under segment number 30 have slightly higher download times due to the exploration of reinforcement learning. However, as time passes, the download time decreases, and the deviation becomes smaller.

Figure 8 shows the performance of *exploration*. The proposed scheduler improves performance by performing a random action if there is enough data in the client buffer. The solid blue line is the result of performing random actions, and the dashed orange line is the result of using the learned optimal policy without any random actions. If a random action is performed, the chunk download time may increase, but if the optimal policy is used, the download time becomes relatively stable.

### 5.3. Network with Packet Loss and Similar RTT Paths

The network environment for the second experiment was configured as shown in Figure 9. There is a fast path with a bandwidth of 50 Mbps and a delay of 6 ms, and a slow path of 5 Mbps and 8 ms. The slow path includes high packet loss rates. In particular, since the overall delay is reduced, the router queue size is set to 100, which is twice the size of the previous experiment. The purpose of this experiment is to see if it can respond well when there is a path with similar RTT but a large loss.

Figure 10a,b studies the chunk download times for all segments and segments after 150 s. In this case, the proposed reinforcement learning scheduler shows better results than other schedulers. This is because, compared to other schedulers where packet loss is not taken into account, the reinforcement learning scheduler can learn an unknown local network by itself.

### 5.4. Chunk Manager

This experiment assumes a general network environment. This section covers the performance of the chunk manager and changes in the epsilon value. Details are given in Figure 11.

Accurately estimating the download time of chunks in the chunk manager can improve the performance of the reinforcement learning scheduler. Figure 12 is a graph comparing the actual chunk download time measured from the client and the estimated chunk download time from the server. We can observe that the estimated download time from the server and the actual download time were measured very closely. However, in most cases, the download time measured by the server has a slightly lower value than the actual download time measured by the client. The reason is that it does not include the time when the client sends an HTTP request to the server. In most cases, the client has a certain error because it uses an empty fast path.

Figure 13 is an epsilon graph. The epsilon is the probability of performing a random action. This value starts at 1 and decreases over time. Figure 13b is the epsilon graph modified by the proposed scheduler. When there is enough video data in the client buffer, the epsilon value is artificially increased to perform a random action. If this value exceeds 1, it means that a random action is performed with only 100% probability.

Figure 14 is a graph showing how many records are accumulated in the replay buffer over time. After learning all 300 chunks, about 1300 scheduling records are stored.

### 5.5. Adaptive Video Bitrate

In this experiment, as shown in Figure 15, it is a network environment in which the bandwidth of the two paths is similar, but the delay difference is large. Unlike previous experiments, the video streaming server provides a total of 3 video bitrates of 1 Mbps, 5 Mbps, and 10 Mbps, with the purpose of checking whether the client can achieve a high video bitrate.

Table 2 shows the average video bitrate for each scheduler. Peekaboo’s performance tends to decrease gradually. ECF still shows good performance. The performance of the proposed reinforcement learning scheduler also gradually increases with time, showing that it exceeds the ECF after 150 s.

Table 3 shows that performance may be degraded if chunks that have performed random actions are included in the client’s video bitrate calculation. Even if we include chunks with random actions, The client can still achieve high video bitrates but cannot increase the video bitrate quickly. The reason is that those chunks cause clients to underestimate network performance.

## 6. Discussion

In this paper, we chose the DQN algorithm for reinforcement learning on the following basis. First, we need to use a model-free algorithm that can operate even when only partial network parameters are known. Second, since there are opportunities for exploration in DASH scenarios, we needed to choose an algorithm that supports off-policy. An on-policy algorithm uses the learned policy to obtain the output and uses this future learning. On the other hand, the off-policy algorithm can use the output that has not been used in the past policy, thus enabling the exploration of new actions when the DASH buffer has enough space. Q-learning uses this off-policy algorithm, and DQN is a Q-learning-based algorithm that uses the replay buffer that performs learning by using data resulting from the previous policy. Third, the action used in the system architecture is a discrete action, i.e., it does not need a policy-based gradient or an actor-critic-based algorithm. There are some continuous actions, but this does not affect the action space. Fourth, DQN’s experience replay is a continuously learning algorithm using a min-batch-sized random sampling. Using random sampling, data can be reused for learning. Finally, most multipath schedulers that use reinforcement learning [20,25] use the DQN algorithm due to the advantages stated above.

## 7. Conclusions and Future Work

In this paper, we proposed a novel MPQUIC scheduler based on reinforcement learning using DQN to improve multimedia streaming performance. We took into account both delay and throughput as a reward for each episode to reduce the video chunk download time. We also proposed a chunk manager so that the reinforcement learning agent can be better informed of the video chunk’s transmissions. We also tuned the hyperparameters for a better balance between exploitation and exploration. We gave evaluation results based on real Linux kernel implementation and Mininet experiments. Our evaluation results show that our proposal outperforms legacy schedulers.

Due to the nature of reinforcement learning, the user’s experience may be degraded by selecting the wrong action in the initial stage. First, there is a way to reuse the results of reinforcement learning when playing the next video. In this case, the initial buffering time can be reduced by using a policy that optimizes the chunk download time when a new video is played. Clearly, the initial buffering time may increase for the first video. Therefore, there may be a way to make minRTT or redundant operation performed in a situation where environmental data is insufficient and to improve policy through random action when there is room in the buffer. We leave this for future work.

## Figures and Tables

**Figure 1 sensors-22-06333-f001:**
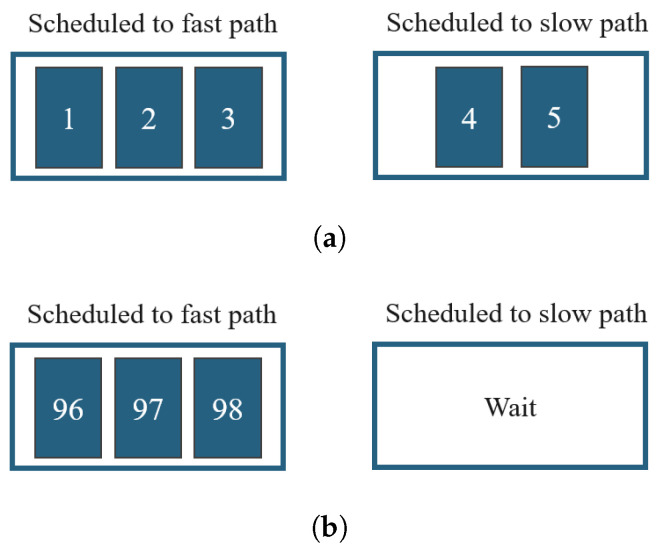
Optimal scheduling method. (**a**) Time 1. (**b**) Time n − 1. (**c**) Time n.

**Figure 2 sensors-22-06333-f002:**
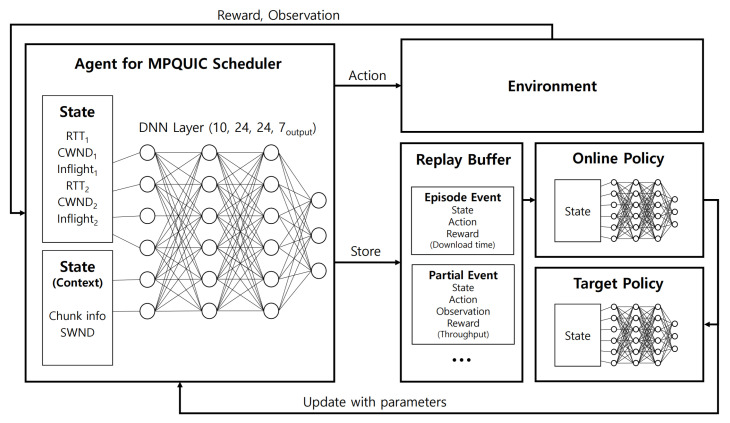
Deep-Q network (DQN) Architecture.

**Figure 3 sensors-22-06333-f003:**
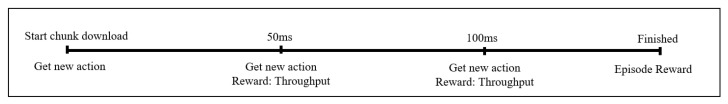
An example of scheduling packets in a chunk, which is equivalent to an episode. The scheduling decision is made every 50 ms.

**Figure 4 sensors-22-06333-f004:**
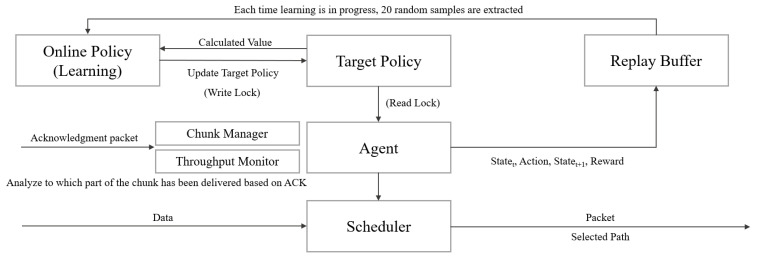
Overall system structure.

**Figure 5 sensors-22-06333-f005:**
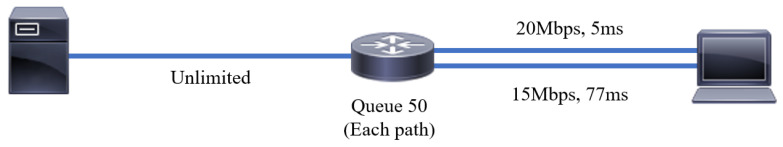
Network configuration with diverse RTT paths.

**Figure 6 sensors-22-06333-f006:**
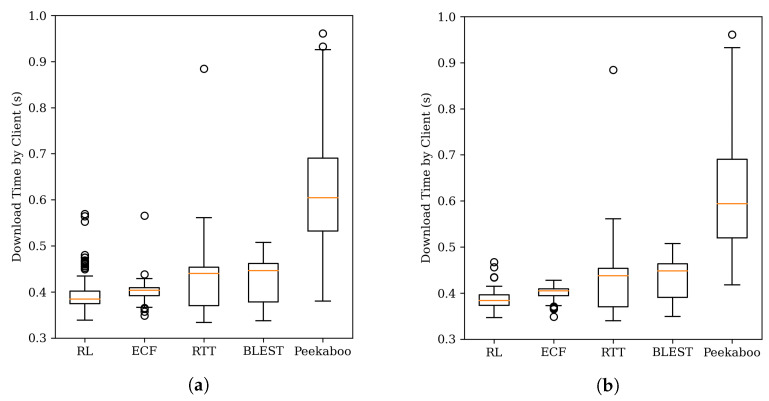
Chunk download times for static network. (**a**) All segments. (**b**) Segment after 150 s.

**Figure 7 sensors-22-06333-f007:**
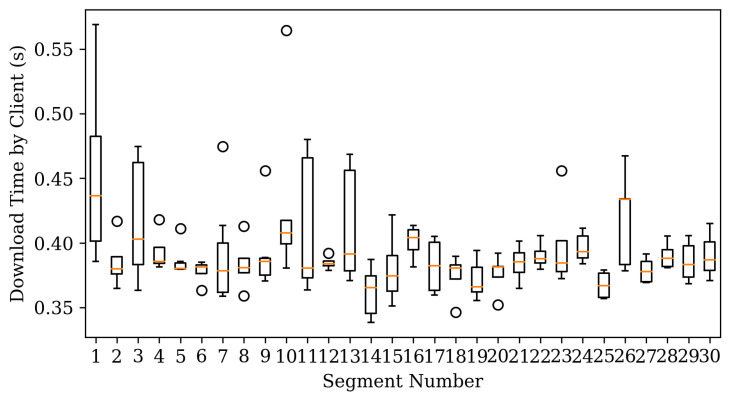
RL scheduler performance change over time.

**Figure 8 sensors-22-06333-f008:**
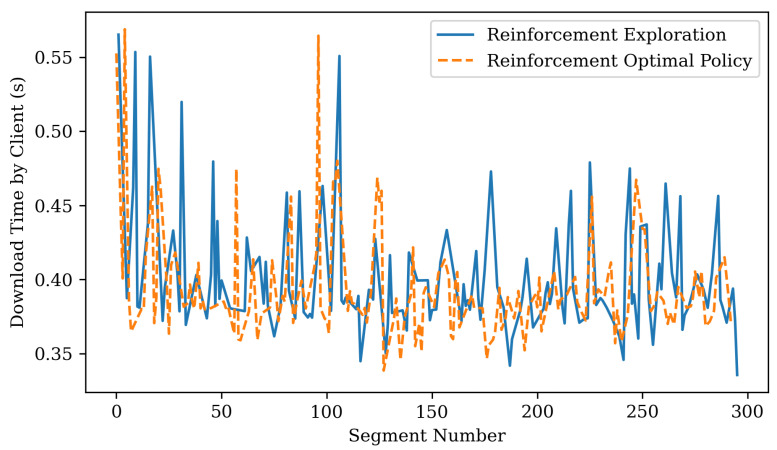
Changes in download time according to policy.

**Figure 9 sensors-22-06333-f009:**
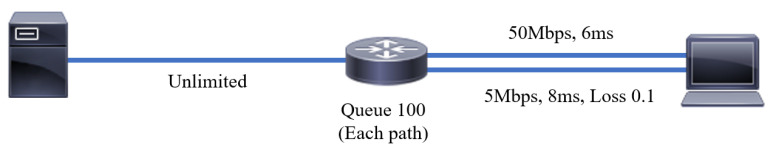
Network configuration with similar RTT paths and packet loss.

**Figure 10 sensors-22-06333-f010:**
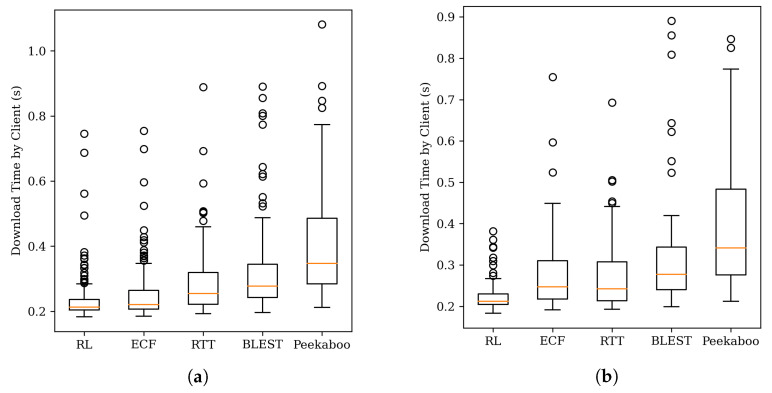
Chunk download times for packet loss network. (**a**) All segments. (**b**) Segment after 150 s.

**Figure 11 sensors-22-06333-f011:**
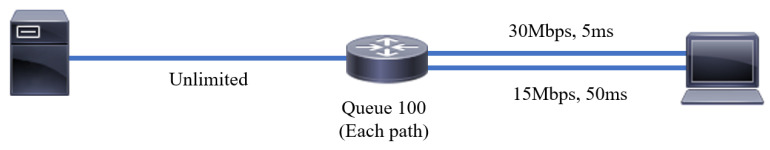
Network configuration to evaluate the chunk manager.

**Figure 12 sensors-22-06333-f012:**
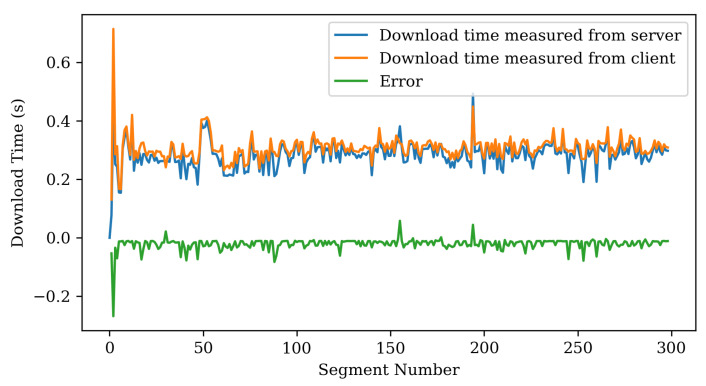
Measurement error between server and client regarding download time in the Chunk Manager.

**Figure 13 sensors-22-06333-f013:**
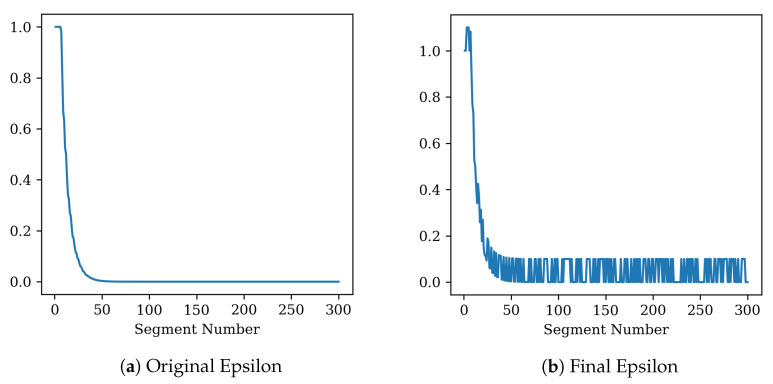
Epsilon. This means the probability that the agent performs a random action. (**a**) is the default epsilon, and (**b**) is the epsilon that responds according to the buffer status of the client.

**Figure 14 sensors-22-06333-f014:**
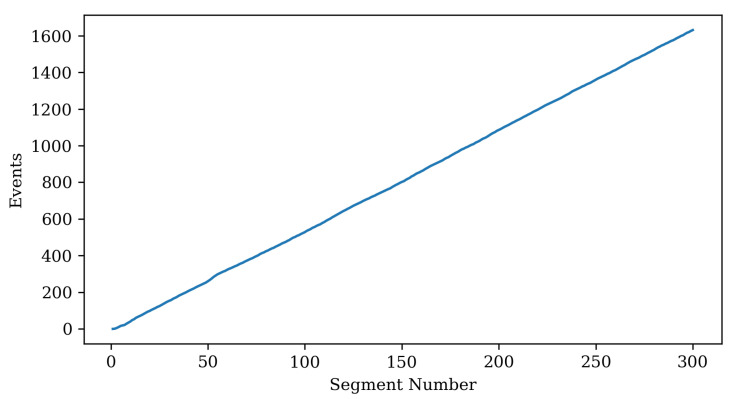
Replay buffer size. An event is a tuple stored in the replay buffer. It has state, performed action, reward, and changed state (=observation).

**Figure 15 sensors-22-06333-f015:**
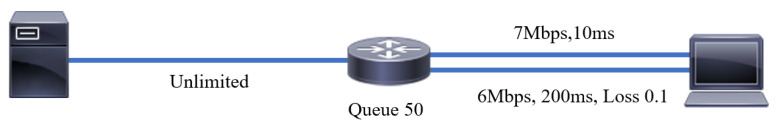
Network configuration to evaluate the performance of adaptive bitrate.

**Table 1 sensors-22-06333-t001:** Example network configuration.

Path	Delay	Congestion Window	Packet Loss Rate
1	5 ms	3	0%
2	50 ms	2	0%

**Table 2 sensors-22-06333-t002:** Average video bitrate for each scheduler. In detail, it is the average of the data corresponding to the left segment range.

Segments	MinRTT	BLEST	ECF	Peekaboo	RL
0–300	4.8 Mbps	4.6 Mbps	7.38 Mbps	2.55 Mbps	5.95 Mbps
50–300	5.0 Mbps	4.9 Mbps	7.49 Mbps	2.48 Mbps	6.55 Mbps
100–300	5.0 Mbps	5.0 Mbps	7.48 Mbps	2.5 Mbps	7.13 Mbps
150–300	5.0 Mbps	5.0 Mbps	7.48 Mbps	2.37 Mbps	7.51 Mbps
200–300	5.0 Mbps	5.0 Mbps	7.47 Mbps	2.06 Mbps	7.52 Mbps
250–300	5.0 Mbps	5.0 Mbps	7.45 Mbps	1.78 Mbps	7.54 Mbps

**Table 3 sensors-22-06333-t003:** Average video bitrate for the proposed scheduler. The left column is the result of the reinforcement learning scheduler without any modifications. The right column is the result of applying the correction for the underestimation of the network.

Segments	RL (Include Chunks in Bitrate Calculations)	RL (Final Version)
0–300	5.11 Mbps	5.95 Mbps
50–300	5.54 Mbps	6.55 Mbps
100–300	6.14 Mbps	7.13 Mbps
150–300	6.52 Mbps	7.51 Mbps
200–300	6.98 Mbps	7.52 Mbps
250–300	7.54 Mbps	7.54 Mbps

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
