# Peer review of "Reinforcement Learning Based Multipath QUIC Scheduler for Multimedia Streaming"

_sensors, 2022, doi:10.3390/s22176333_

Round 1
Reviewer 1 Report
This paper proposed reinforcement learning to achieve a low video chunk download time. The authors propose an MPQUIC scheduler to improve the performance of video 54 streaming via a novel deep reinforcement learning algorithm that uses Deep Q-Network 55 (DQN). The authors implement the new scheduler on the Linux kernel, and give results using the Mininet experiments. The work is interesting, and worth to be published. However, some question should be answered before publication.
1)The reinforcement learning method used in this paper is general, the authors should clarify the novelty of this work.
2)The author should clarify how to training the data by using the deep learning method.
3)What kind of feature is extracted for training?
4)The diagram of the deep Q-Network 55 used in this work should sketched and explained in detailed.
Reviewer 2 Report
The paper is an interesting project application problem, but there is still much room for theoretical discussion to improve.
1. In the introduction, why the DQN algorithm proposed in the paper is needed after the engineering problems analyzed;
2. In reinforcement learning, there is more than one excellent algorithm DQN. Why did the author choose DQN? authors should explain it.
3. Some of the algorithm important parameters have not been expanded to explain why such a value is set.
4. At the end of the paper, there is no discussion section;
5. "Sensors" is an excellent journal. Please carefully study and quote recent excellent papers related to this article.
Reviewer 3 Report
I wonder whether this article can fit the general scope.
I cannot see anything related to sensors or even provide similar wordings in the article.
Some sentences are not clearly written. “…. most schedulers attempt to optimize either latency or throughput”, .. .” MinRTT [3] can achieve high throughput, but is not optimized in terms of latency”
The sentences made me very confusing. Further elaboration on the optimization should be provided. Similar works on AI methods should be provided.
The article proposes the general reinforcement learning approach. What is the novelty of this research?
The articles have many sections, some of them can be combined or better organized.
Why DQN is used? There is no comparison of results with other algorithms provided.
How the evaluation was performed and the settings are not clear.
Conclusions and future works should be elaborated.
English proofreading is required.
Round 2
Reviewer 1 Report
All my question are considered and addressed. I recommend that the paper is ready for publication.
Reviewer 2 Report
Authors have addressed my conerns, it can be published
Reviewer 3 Report
Comment addressed.